# Logic Agent: Enhancing Logical Reasoning for Large Language Models with Logic Rules

## Abstract

Chain-of-Thought (CoT) prompting has become a key strategy for enhancing the inferential abilities of large language models (LLMs) in reasoning tasks. However, it often struggles with ensuring reasoning validity and maintaining informativeness. This paper presents the Logic Agent (LA), a novel framework designed to boost the validity of reasoning in LLMs through strategic logic function calls. Distinct from traditional methods, LA converts LLMs into dynamic agents that apply propositional logic rules, transforming natural language inputs into structured logical forms. The agent utilizes a robust suite of predefined functions to guide the reasoning process effectively. This approach can enhance the structured and coherent generation of reasoning outputs, improving their interpretability and logical consistency. Through detailed experiments, we showcase LA's ability to adapt across different LLM sizes, significantly enhancing the accuracy of complex reasoning tasks across various domains.

## 1 Introduction

The quest for augmenting the reasoning capabilities of language models has been a focal point of recent advancements in the evolving landscape of artificial intelligence. Chain-of-Thought (CoT) prompting Wei et al. (2022); Kojima et al. (2022); Wang et al. (2022b); Chu et al. (2023) marked a significant stride in this journey, revealing the potential of large language models (LLMs) to mimic human-like reasoning processes. These advancements have led to remarkable achievements, with LLMs demonstrating proficiency in a variety of competitive examinations, including those focused on mathematics Li et al. (2022); He-Yueya et al. (2023); Imani et al. (2023) and reading comprehension Wang et al. (2023); Xiao et al. (2023).

However, despite its implications in various reasoning tasks, CoT has faced limitations, particularly in validating reasoning and ensuring the informativeness of its outputs. Lanham et al. (2023) Their performance in logical reasoning tasks, a critical component of examinations like the Law School Admission Test (LSAT) and Chinese civil service selection exams, remains notably inferior to that of well-trained humans liu et al. (2023).

Figure 1 shows an example of such questions. Crafted by experts to challenge human logical reasoning abilities, they require a valid and rule-bound chain of logic that is often non-trivial to discern. Testees must engage in abstract thinking, translating contexts into logical symbols and applying strict inference rules to form logical chains. This gap highlights a critical challenge: the ability of LLMs to consistently follow rules and verify the validity of logic chains. As illustrated in Figure 1, a GPT-4 model struggles with the deduction of the contrapositive law, despite having conceptual knowledge of it. One reason can be that there is no strict guarantee for a statistical system such as LLM to ensure correct complex reasoning chains across contexts.

Inspired by the integration of the neural network with formal symbolic solvers Azerbayev et al. (2023); Jiang et al. (2023); Thakur et al. (2023), tool-use Gao et al. (2023); Schick et al. (2023); Paranjape et al. (2023), and constrained decoding Geng et al. (2024), we address this issue by introducing Logic Agent (LA), an agent-based constrained generation framework, leveraging propositional logic and inference rules as fundamental guides to constructing logically sound inference chains. LA is designed to steer Large Language Models (LLMs) toward a trajectory of enhanced logical coherence and interpretability by introducing symbolic reasoning. In particular, we let an LLM serve as a decision-making agent and make a callable symbolic reasoning agent by assembling a set of essential

**Context:**
**If** the Moon's surface was once a magma ocean, **then** the distribution of many elements on it should be continuous. The magma ocean holds the key to unlocking the mystery of the Moon's origin. **If** the existence of a magma ocean is confirmed, **then** the 'Giant Impact Hypothesis' becomes the most plausible explanation for the Moon's origin.

**Question:**
From this, we can infer:

**Options:**
A. If the Moon's surface was never a magma ocean, then the distribution of elements on its surface is not continuous.
B. If the 'Giant Impact Hypothesis' is not the most plausible explanation for the Moon's origin, then it indicates that the distribution of elements on the Moon's surface is not continuous.
C. If the distribution of elements on the Moon's surface is not continuous, then the Moon's surface was never a magma ocean.
D. If the distribution of elements on the Moon's surface is continuous, then the 'Giant Impact Hypothesis' will become the most plausible.

- - - - - - - - - - - - - - - - - - - - - - - - - - - - - - - - - - - - -

**Atoms:**
P: Moon's surface was a magma ocean
Q: Continuous distribution of elements
R: the 'Giant Impact Hypothesis' is the most plausible explanation

**Implies:**
$P \rightarrow Q$
$P \rightarrow R$

**External guides:**
$\neg Q \rightarrow \neg P$
$\neg R \rightarrow \neg P$    [Contrapositive Law]

$\neg P \rightarrow \neg Q$
$\neg P \rightarrow \neg R$    [Invalid]

**GPT-4 CoT Freeform:**
...
Now, let's translate each of the given inferences:
A. $\neg P \rightarrow \neg Q$

This is the contrapositive of the first part of the original statement ($P \rightarrow Q$), so it is logically equivalent and true. ✗
...

**GPT-4 LA Guided:**
...
Given the original statement ($P \rightarrow Q$), according to Contrapositive($P \rightarrow Q$): $\neg Q \rightarrow \neg P$, so the inference in option A is invalid. ✓
...

Figure 1: An example of logical reasoning problems in competitive exams. GPT-4 can handle abstract logical reasoning, however, it fails to conduct a valid inference chain.

formal logical rules. The LLM agent is taught to make use of the symbolic reasoning agent in its instructions so that formal reasoning steps can be guaranteed strictly correct.

With LA, LLMs are guided towards a path of logical coherence and interpretability. We first define the essentials of compositional logic, i.e. the logic components and syntax. This step serves as the initial step, converting complex natural language statements into structured compositional logic representations. Second, we define the functions for applying deduction rules, given a logic expression, we are able to form an inference chain with implicit logic. These functions are tools for LLMs to use. Lastly, we prompt LLMs to decide which rule to apply in different states. When LLMs call a rule, the output of the corresponding function is guaranteed to give valid logic chains for LLMs to make judgments on the truthfulness of the hypotheses.

In our study, we rigorously evaluated the Logic Agent (LA) framework using a mix of commercial and open-source Large Language Models, including OpenAI's GPT-4 and various Hugging Face models. Our findings, across this diverse range of models, consistently highlight LA's effectiveness in enhancing logical reasoning in complex tasks. Alongside our experimental insights, we're releasing our code to contribute to ongoing research. To the best of our knowledge, this is the first initiative to integrate propositional logic into LLMs at such a scale.

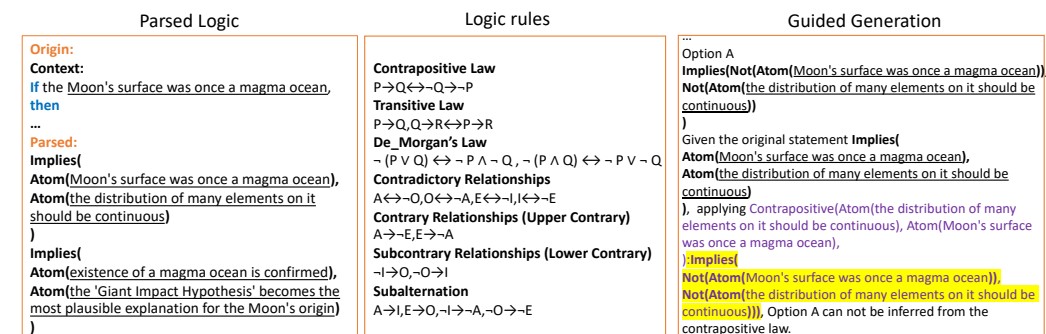

Figure 2: The LA framework. Highlighted texts are the output of pre-defined functions.

## 2 RELATED WORK

Traditional pre-trained models have primarily tackled logical reasoning through statistical training, a connectionist approach that often misinterprets the complexity of language. Similarly, formal symbolic systems, while precise, struggle with the adaptability needed for diverse linguistic phenomena. This backdrop sets the stage for the introduction of new approaches to complex reasoning in Large Language Models (LLMs).

**Reasoning Paradigms in Large Language Model Prompting:** The development of few-shot Wei et al. (2022) and zero-shot Kojima et al. (2022) Chain-of-Thought prompting has been instrumental in enabling LLMs to tackle complex reasoning tasks. Subsequent developments have introduced varied data structures, such as Tree-of-Thought Yao et al. (2023), Graph-of-Thought Besta et al. (2024), and Program-of-Thought Chen et al. (2022), enhancing LLMs' capabilities to reflect on and evaluate their reasoning processes. Moving beyond basic prompting strategies, the ReAct model Yao et al. (2022) intertwines reasoning with actionable tasks like search, while the Selection-Inference framework Creswell et al. (2023) employs a two-step process of context formation and logical chaining. Although these approaches parallel ours in process structure, they do not incorporate explicit logical rules, and the chaining mechanism is entirely model-dependent. The use of external tools within prompting paradigms, particularly for tasks necessitating additional knowledge, represents another significant advancement. In mathematical reasoning, tools such as calculators have proven invaluable. Analogously, in our methodology, predefined functions for applying inference rules are akin to external, a concept previously unexplored in this context. Another paradigm shift in LLM prompting is the division of complex tasks into subproblems or the collaborative engagement of diverse models. Cumulative reasoning Zhang et al. (2023) adopts a streamlined, iterative approach utilizing distinct LLMs as AI agents; ScratchPad Nye et al. (2021) contributes to multi-step reasoning by revealing intermediate steps; Meta-prompting Suzgun & Kalai (2024) envisions LLMs as orchestrators in a collaborative environment, responsible for decomposing complex tasks, delegating sub-tasks to specialized models, facilitating inter-model communication, and applying critical analysis throughout. Our approach similarly harnesses the LLMs' decision-making capability in selecting appropriate inference rules, aligning with this broader trend of utilizing LLMs for complex, collaborative reasoning processes. Unlike previous attempts, we leverage the computational power and contextual understanding of LLMs to act as agents that dynamically invoke logic rules. This integration enables the LLMs to not only process language with their inherent sophistication but also apply logical reasoning in a structured and accurate manner, akin to utilizing a calculator for mathematical enhancements. Apart from that, recent studies have explored instruct-tuning Large Language Models (LLMs) with specific datasets to enhance their abstract reasoning capabilities. LogiCoT Liu et al. (2023) fine-tunes an LLAMA-7B model using logical chaining data, demonstrating significant improvements across various logical reasoning tasks; LogicLLM Jiao et al. (2023) employs a self-supervised post-training approach tailored for logical reasoning enhancements; Symbol-LLM Xu et al. (2023) leverages symbolic data within a two-stage tuning framework to imbue a LLAMA-2-CHAT model with symbolic knowledge. While these approaches underscore the potential of fine-tuning strategies in augmenting LLMs, our work distinguishes itself as the first to specifically address and enhance logical reasoning capabilities at the decoding stage, employing a multi-agent strategy to elevate the process.

**Formal Reasoning:** Formal reasoning systems have primarily been developed to address mathematical challenges. Peano Poesia & Goodman (2023), designed to solve educational mathematical problems, employs dependent types to encode mathematical definitions and proofs, echoing the structured approach in our work. Yet, our focus diverges towards logical reasoning scenarios, an area where systems like Peano have traditionally been less potent. Addressing formal logical reasoning, LINC Olausson et al. (2023) leverages LLMs as FOL language translators to attain formal representations of contextual information, complemented by traditional theorem provers for validation. LINC's approach, which employs a voting strategy to resolve inconsistencies in FOL language generation, contrasts with our method which adopts a more flexible propositional logic to distill the abstract essence of context while meticulously controlling the validity of generative reasoning. Furthermore, the exploration of language models as theorem provers has introduced systems like LangPro Abzianidze (2017), a natural language theorem prover that harnesses higher-order logic to assess linguistic expressions' consistency. LangPro's reliance on CCG parsing and a dedicated knowledge base for generating Lambda Logical Forms (LLFs) presents a contrast to our work, which utilizes propositional logic, thereby circumventing the need for a theorem-proving knowledge base. In parallel, semantic-constrained decoding techniques, as exemplified by NEUROLOGIC DECODING Lu et al. (2020), enable language models to generate contextually coherent text while adhering to complex lexical constraints. Our approach resonates with this paradigm, albeit with a distinct focus on employing constrained generation paired with guided deduction rules, thereby carving a unique niche in the landscape of formal reasoning and logical inference.

## 3 LOGIC AGENT

Distinctively, we encapsulate the logical reasoning process into callable function forms, packaging logic rules as tools for LLM agents. This strategic shift in leveraging LLMs as autonomous decision-makers, equipped with a toolkit of generalized logic reasoning functions, marks a significant departure from existing models.

Figure 2 presents the Logic Agent (LA) framework's architecture. Initially, natural language inputs undergo logic parsing on the left, resulting in structured logic forms (see Section 3.1). The center highlights the application of deduction rules for logical inference (see Section 3.2). Finally, on the right, the constrained generation process employs these inferences to produce contextually relevant and logically coherent outputs, illustrating the LA's systematic approach to enhancing reasoning in large language models (see Section 3.3). At the heart of LA lies the meticulous definition and utilization of compositional logic essentials, encompassing both the critical logic components and their associated syntax. This pivotal initial step involves the intricate transformation of complex natural language statements into structured representations of compositional logic.

### 3.1 LOGICAL CONSTRUCT CLASSES

Within LA, various classes of logical constructs are parsed and utilized. These include:

`Variable`: Represents a variable symbol in logical expressions. `Atom`: Denotes an atomic formula, the fundamental unit of logical statements. `Not`: Embodies the negation operation in logic. `And`: Indicates logical conjunction, combining multiple propositions. `Or`: Symbolizes logical disjunction, offering alternative propositions. `Implies`: Represents the implication relationship between propositions. `Equiv`: Denotes logical equivalence between statements. `Exists` and `Forall`: Represent existential and universal quantification, respectively, allowing for the expression of propositions about 'some' or 'all' entities within a domain. Rule-based functions within LA parse these logical constructs and quantified sentences, ensuring accurate representation and manipulation of logical expressions.

### 3.2 INFERENCE RULES

On this foundational layer, LA incorporates a suite of defined functions for applying various deduction rules. These functions serve as advanced tools for LLMs, facilitating the formation of inference chains that integrate both explicit and implicit logic elements. This enables LLMs to navigate the complexities of logical deduction, maintaining structured and coherent reasoning throughout.

The key inference rules and their corresponding functions in LA include:

| Dataset | Size | Target |
|---|---|---|
| ReClor dev | 500 | 4-way multi-choice |
| AR-LSAT test | 230 | 5-way multi-choice |
| LogiQA22 | 1,354 | 4-way multi-choice |
| ConTRoL test | 805 | E, C, N |
| NaN-NLI test | 259 | E, C, N |
| RuleTaker dev | 10,068 | Yes, No |
| ProofWriter dev | 10,158 | Yes, No |

Table 1: The statistics of the datasets. ("E" refers to "entailment"; "C" refers to "contradiction"; "N" refers to "neutral".)

`Contrapositive`: A function applying the contrapositive law, turning implications into their logically equivalent forms. `Transitive`: A function for the transitive law, linking propositions through a common term. `De_Morgans`: Implements De Morgan's laws, transforming conjunctions and disjunctions while preserving logical equivalence.

We also integrate the foundational principles of categorical propositions, which is essential to syllogistic logic. There are four key proposition types: `SAP (A)` - Universal Affirmative, `SIP (I)` - Particular Affirmative, `SEP (E)` - Universal Negative, and `SOP (O)` - Particular Negative. Below are the corresponding functions:

`Contradictory`: A function handling contradictory relationships, identifying mutually exclusive propositions. `Contrary`: Manages contrary relationships, where two propositions cannot be true simultaneously but can be false together. `Subcontrary`: Deals with subcontrary relationships, where two propositions cannot be false simultaneously but can be true together. `Subalternation_forward` and `Subalternation_backward`: Functions facilitating subalternation, capturing the inferential relationships between universal and particular propositions. Through these specialized functions, LA empowers LLMs to apply logical reasoning accurately and effectively, enhancing their capability to tackle complex reasoning tasks with a higher degree of precision and reliability.

### 3.3 RULE-GUIDED GENERATION

We prompt LLMs to discern and decide upon the most appropriate rule to apply in varying states of reasoning. This dynamic interaction empowers LLMs to judiciously invoke the corresponding functions, each meticulously crafted to guarantee the generation of valid logic chains. Consequently, LLMs are equipped with a powerful mechanism to scrutinize the veracity of hypotheses, making informed judgments based on the logically consistent chains produced. We use in-context examples to demonstrate how these functions are called in the guided generation process and leverage the capabilities of existing LLMs developed by OpenAI and HuggingFace. These models offer a robust starting point, owing to their advanced language understanding and processing abilities. However, our approach goes beyond the conventional use of LLMs by optimizing each component for its specific role in the logical reasoning process. This targeted optimization is key to transcending the current limitations of LLMs in handling the nuanced and rule-bound nature of logical reasoning tests.

By integrating a structured, rule-guided reasoning methodology into the operational framework of LLMs, LA aims to improve not only the logical precision of these models but also their interpretability and coherence. The incorporation of propositional logic, deduction rules, and a strategic prompting mechanism positions LA as an innovative approach. It seeks to bridge the current divide between the computational efficiency of LLMs and the detailed, logical discernment typical of human reasoning.

### 3.4 TASKS

We consider various logical reasoning tasks, including Multi-Choice Reading Comprehension (MCRC), Natural Language Inference (NLI), and True-or-False questions (TF).

The datasets we use are listed in Table 1. ReClor Yu et al. (2020), AR-LSAT Wang et al. (2022a), and LogiQA22 liu et al. (2023) are three renowned multi-choice reading comprehension datasets for logical reasoning. ReClor and AR-LSAT are collected from verbal reasoning questions in competitive tests like the LSAT (Law School Admission Test) exam. LogiQA22 is collected from the Chinese Civil Service Examination in the year 2022. ConTRoL Liu et al. (2021) and NaN-NLI Truong et al.

| Task | MCRC | | | NLI | | TF | |
|------|--------|---------|----------|---------|----------|----------|-------------|
| **Dataset** | **Reclor** | **AR-LSAT** | **LogiQA22** | **ConTRoL** | **NaN-NLI** | **RuleTaker** | **ProofWriter** |
| **Human avg.** | 63.00 | 56.00 | 83.00 | 87.00 | 94.00 | 84.00 | 82.00 |
| **Human Ceiling** | 100.00 | 91.00 | 99.00 | 94.00 | 100.00 | 95.00 | 93.00 |
| **GPT-3.5-Direct** | 56.28 | 51.31 | 41.14 | 57.94 | 56.86 | 55.33 | 54.68 |
| **GPT-3.5-CoT** | 56.90 | 51.45 | 42.92 | 58.29 | 55.54 | 55.88 | 53.02 |
| **GPT-3.5-LA** | 59.73 | 55.29 | 42.98 | 62.01 | 61.34 | 71.30 | 73.85 |
| **GPT-4-Direct** | 88.54 | 74.21 | 60.11 | 56.34 | 77.07 | 59.85 | 61.58 |
| **GPT-4-CoT** | 89.06 | 73.49 | 58.43 | 56.97 | 77.83 | 61.43 | 60.64 |
| **GPT-4-LA** | 89.47 | 77.28 | 60.67 | 58.93 | 80.66 | 65.84 | 68.42 |
| **Davinci-002-Direct** | 20.41 | 13.54 | 11.02 | 8.43 | 10.78 | 25.98 | 22.54 |
| **Davinci-002-CoT** | 19.43 | 18.85 | 13.27 | 13.61 | 15.34 | 26.84 | 27.33 |
| **Davinci-002-LA** | 27.45 | 22.60 | 30.68 | 15.58 | 24.73 | 32.10 | 33.54 |
| **LLaMA-2-Direct** | 17.31 | 12.70 | 18.55 | 20.12 | 22.08 | 25.50 | 23.39 |
| **LLaMA-2-CoT** | 15.62 | 13.76 | 16.03 | 21.75 | 25.44 | 22.39 | 23.16 |
| **LLaMA-2-LA** | 23.76 | 21.63 | 30.21 | 25.48 | 22.76 | 28.79 | 25.11 |
| **Mixtral-8x7b-Direct** | 48.92 | 41.40 | 38.97 | 50.84 | 50.13 | 46.84 | 44.80 |
| **Mixtral-8x7b-CoT** | 49.21 | 44.33 | 40.96 | 50.32 | 53.04 | 48.52 | 45.85 |
| **Mixtral-8x7b-LA** | 50.58 | 45.95 | 44.92 | 52.25 | 55.96 | 52.53 | 55.68 |

Table 2: Main results. All results are in %.

(2022) are two logical reasoning datasets for the natural language inference task. The task is to decide whether a hypothesis can be logically entailed by the premises. ConTRoL features entailment relationships for long texts, and NaN-NLI is for negations. Both datasets are three-way classification tasks. RuleTaker Clark et al. (2020) and ProofWriter Tafjord et al. (2020) are two synthetic datasets widely used in formal logic reasoning. They take the form of yes-or-no questions, which are designed to test the ability of models to understand and apply rules and facts stated in natural language.

The tasks are evaluated with few-shot prompting, we use three in-context examples, covering different inference rule scenarios. For the implementation, we use a series of models from the OpenAI suite, including DAVINCI-002, GPT-3.5-TURBO, and GPT-4. DAVINCI-002 is the GPT base model currently supported by OpenAI API. GPT-3.5-TURBO and GPT-4 are two chat models available in the OpenAI API. Furthermore, we extend our evaluation to Huggingface models like LLAMA-2-13B Touvron et al. (2023) and MIXTRAL-8X7B-V0.1 Jiang et al. (2024), thereby encompassing a broad spectrum of AI models. LLAMA-2-13B is a 13B open LLM developed by Meta. MIXTRAL-8X7B-V0.1 is a Mixture-of-Expert (MoE) model developed by MistralAI. This diverse selection includes both base and instruction-tuned models, covering a range of open-source and closed-source options, to provide a comprehensive overview of the capabilities and performance variations across different AI architectures in logical reasoning tasks. We use the guidance library [1] for implementing our rule-constrained generation framework.

# 4 EXPERIMENTS

We use a diverse range of datasets and models to ensure a robust and thorough assessment of our framework. We detail our experimental setup, the metrics used for evaluation, and our main findings.

## 4.1 EXPERIMENTAL SETUP

**Baselines**: Our experimental baselines comprise two distinct approaches: direct answering and Chain-of-Thought (CoT) reasoning. To facilitate a fair comparison between base models and instruction-tuned models, we provide three in-context examples for both the direct answering and the CoT scenarios. This approach aids LLMs in generating answers that can be directly compared with the gold labels.

**Data preprocessing**

- For the Multiple-Choice Reading Comprehension (MCRC) task, we combine the context, question, and options to form a single input.

---

[1]https://github.com/guidance-ai/guidance

- In Natural Language Inference tasks, premises and hypotheses are concatenated, with a distinct identifier prefacing each segment.
- For True-or-False questions, we concatenate the context with the question to generate a cohesive input prompt.

**Metrics** To assess the performance of LLMs in our experiments, we employ the *exact-match* metric. This involves prompting LLMs to generate answers either as the first token (direct answer) or at the end of the generation process (CoT and LA). The extracted answers are then compared with the gold labels to calculate the accuracy score.

## 4.2   RESULTS

The primary outcomes of our experiments are summarized in Table 2, where we juxtapose the performances of different models under various logical reasoning tasks. These tasks span multiple-choice reading comprehension (MCRC), natural language inference (NLI), and true-or-false (TF) questions, utilizing datasets such as ReClor, AR-LSAT, and LogiQA22 for MCRC, ConTRoL and NaN-NLI for NLI, and RuleTaker and ProofWriter for TF tasks. The human performance benchmarks, as referenced in the table, are sourced from prior research Yu et al. (2020); Wang et al. (2022a); liu et al. (2023).

**Direct Answer vs. Chain-of-Thought (CoT)**: Our analysis reveals that, in the context of the logical reasoning tasks tested, the few-shot CoT approach marginally outperforms the direct answer methodology. However, this superiority is not uniform across all cases. In certain instances, the CoT method appears to detrimentally impact the overall results, suggesting limitations in the effectiveness of CoT prompting in some logical reasoning scenarios. This observation highlights the inherent challenge in using CoT prompting to navigate the complexities of logical reasoning, especially in tasks where intricate inference is required.

**Performance Across Models**: Our further analysis delves into the performance distinctions across various models, highlighting the contrasts between advanced models such as GPT-4 and base models like DAVINCI-002 and LLAMA-2-13B.

DAVINCI-002, as a base model, shows distinct performance characteristics under the LA framework. For instance, in the MCRC task on the ReClor dataset, DAVINCI-002 under LA achieves a 27.45% accuracy, a notable improvement from its Direct answer performance at 20.41%. This trend is consistent across other datasets, such as in LogiQA22, where DAVINCI-002's accuracy increases from 11.02% (Direct) to 30.68% (LA). These results suggest that the structured reasoning provided by LA can significantly enhance the logical reasoning abilities of even base models, enabling them to outperform their standard configurations.

Similarly, LLAMA-2-13B, another base model, exhibits a marked performance enhancement with the application of LA. In the TF task using the RuleTaker dataset, LLAMA-2-13B registers an accuracy of 28.79% under LA, compared to 25.50% in the Direct answering format. In the more challenging ProofWriter dataset, the model improves from 23.39% (Direct) to 25.11% (LA). These improvements, while not as pronounced as those seen with advanced models like GPT-4, nonetheless indicate that LA can elevate the performance of base models in logical reasoning tasks.

Comparatively, advanced models like GPT-4 demonstrate a more significant leap in performance with the LA approach. This is particularly evident in datasets that require complex logical deductions, such as ProofWriter, where GPT-4 with LA achieves a 68.42% accuracy, substantially higher than both its Direct (61.58%) and CoT (60.64%) counterparts.

This comparative analysis across different models underscores the versatility of the LA framework. While advanced models like GPT-4 naturally exhibit higher baseline performances, the introduction of LA leads to substantial improvements in logical reasoning tasks across all model types, including base models like DAVINCI-002 and LLAMA-2-13B. This suggests that LA's structured, rule-guided reasoning approach is universally beneficial, enhancing the logical reasoning capabilities of a wide range of LLMs.

**LA's Efficacy**: The implementation of LA consistently enhances accuracy across various datasets, underscoring its effectiveness in logical reasoning. In the TF tasks using the RuleTaker dataset, LA with GPT-3.5 achieves an impressive 71.30% accuracy, a substantial leap from the 55.33% in the

Direct approach and 55.88% in the CoT approach. Similarly, in the ProofWriter dataset, GPT-3.5 with LA reaches 73.85% accuracy, outperforming both its Direct (54.68%) and CoT (53.02%) formats. These figures highlight LA's capability to significantly refine the reasoning process in LLMs, enabling them to handle complex logic with greater precision and reliability. The improvement is even more pronounced with advanced models like GPT-4, where the accuracy in the RuleTaker dataset jumps to 65.84% under LA, compared to 59.85% (Direct) and 61.43% (CoT). This consistent pattern across various models and datasets firmly establishes LA as a transformative approach in logical reasoning, bridging the gap between computational AI and nuanced human-like reasoning. We present a detailed case study in Appendix **??**. This case study meticulously demonstrates how LA navigates complex logical reasoning tasks, showcasing its capabilities and the enhancements it brings to the decoding stage of Large Language Models (LLMs).

**Task-Specific Insights**: Delving into task-specific performances, we observe that LA aligns exceptionally well with the demands of MCRC and NLI tasks, as evidenced by GPT-4's superior performance in the ReClor and NaN-NLI datasets. The tailored application of LA's rule-based reasoning to each task's unique requirements elucidates its broad applicability and effectiveness. The differential performance uplifts across datasets highlight the adaptability of LA. For instance, the significant accuracy increase in the ProofWriter dataset for GPT-4 underscores LA's capacity to handle datasets requiring complex logical deductions. This adaptability is crucial for tailoring reasoning enhancements to specific task demands.

## 5 Discussion

### 5.1 GPT-4 as Logic Parser

GPT-4, despite its occasional inconsistencies in generating new logical expressions, exhibits a noteworthy capability in parsing natural language into formal logic. This ability is particularly relevant to our Logic Agent (LA) framework, where accurate translation of natural language into propositional logic is crucial.

To harness GPT-4's parsing capabilities, we crafted specific prompts aimed at guiding the model to translate natural language statements into propositional logic forms. These forms are then seamlessly integrated into the deduction functions of LA. A critical requirement for this integration is the compatibility of GPT-4's output with our framework's syntax. Therefore, the prompts are designed not only to elicit the correct logical structures but also to ensure that these structures adhere to the syntax conventions of our default parser.

| Dataset | Default parser | GPT-4 parser |
|---|---|---|
| ReClor dev | 59.73 | 60.65 |
| AR-LSAT test | 55.29 | 55.87 |
| LogiQA22 | 42.98 | 44.07 |
| ConTRoL | 62.01 | 65.24 |
| NaN-NLI | 61.34 | 63.48 |
| RuleTaker dev | 71.30 | 71.45 |
| ProofWriter dev | 73.85 | 72.13 |

Table 3: GPT-3.5-TURBO model results with GPT-4 as the parser.

To evaluate the effectiveness of GPT-4 in this role, we conducted experiments comparing its parsing capabilities with our default logic parser. The comparative results, as detailed in Table 3, demonstrate a slight edge in performance when utilizing GPT-4 as a parser. This finding underscores the efficiency and accuracy of GPT-4 in interpreting and translating complex logical statements from natural language into formal logic constructs.

However, it's important to consider the trade-offs involved. Utilizing GPT-4 as a parser introduces additional computational costs, and there may be instances of variability in the parsing quality. These factors necessitate a careful assessment of the cost-benefit ratio, especially in scenarios where computational resources are a limiting factor or where absolute consistency in logic parsing is critical.

Our findings suggest that while GPT-4 can effectively augment our framework as a neural parser, its integration should be strategically employed, taking into account the specific requirements and constraints of the given logical reasoning task. The potential of GPT-4 to enhance the versatility and

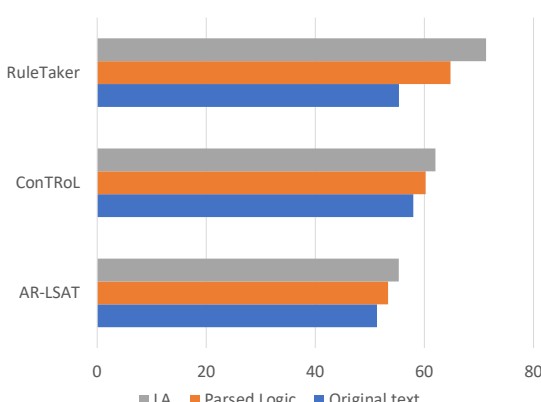

Figure 3: GPT-3.5-TURBO results on ablation test.

adaptability of logical reasoning frameworks is clear, yet its application needs to be tempered with an understanding of its limitations and costs.

## 5.2 ABLATION STUDY

An essential aspect of our research was to ascertain the specific contribution of the parsed logic within the LA method. To achieve this, we conducted an ablation study where we tested the impact of augmenting text with parsed logic on the direct answer approach, while deliberately omitting the constrained generation component integral to LA.

This approach allowed us to isolate and understand the effectiveness of the logic parsing process in isolation. By comparing the performance of models using only parsed logic-augmented text for direct answering with their performance under the full LA framework, we could assess the incremental value added by the constrained generation aspect of LA.

We choose one dataset from each task and use GPT-3.5-TURBO as the tested model. The results are shown in Figure 3. Across the three datasets, we observed a noticeable decrease in accuracy when the models were deprived of the constrained generation process and relied solely on the parsed logic-augmented text. This decline in performance underscores the significance of the constrained generation component in the LA framework. It highlights that while the logic parsing capability is a valuable contributor to the model's overall performance, the full potential of LA is realized only when it is coupled with the sophisticated generation constraints that guide the model towards more logically coherent and accurate conclusions.

## 6 CONCLUSION

In this study, we present Logic Agent (LA), an innovative framework guided by logic rules to enhance the logical reasoning capabilities of Large Language Models (LLMs). Our comprehensive experiments across various models and datasets demonstrate that LA, with its integration of propositional logic and deduction rules, consistently surpasses traditional reasoning approaches. Notably, it shows superior performance in tasks requiring intricate logical deductions, highlighting its potential to bridge the gap between AI computational power and human-like logical reasoning. The exploration of GPT-4 as a neural logic parser further reveals the feasibility and challenges of incorporating advanced LLMs within logical reasoning systems. Looking ahead, the refinement of LA for broader applications and its scalability remain pivotal areas for future research. In sum, the LA framework not only elevates the performance of LLMs in complex reasoning tasks but also paves the way for more sophisticated and interpretable AI reasoning capabilities.

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
