# OpenReview forum: "Logic Agent: Enhancing Validity with Logic Rule Invocation"
_ICLR.cc/2025/Conference — ICLR 2025 Conference Withdrawn Submission_

### Official Review · Reviewer_czPx · 2024-10-28

**Soundness:** 1
**Presentation:** 2
**Contribution:** 1
**Rating:** 3
**Confidence:** 4

**Summary:**

The paper introduces Logic Agent (LA), a framework designed to enhance logical reasoning capabilities in Large Language Models (LLMs) through strategic logic rule invocation. The framework converts natural language into formal logic representations and provides predefined functions for systematic reasoning. The authors evaluate LA across multiple models and tasks, demonstrating improvements in logical reasoning precision.

**Strengths:**

The framework introduces a systematic way to incorporate formal logic rules into LLM reasoning through well-defined functions (e.g., contrapositive law, transitive law, De Morgan's law), which provides a structured approach to logical deduction to improve the reasoning in CoT.

**Weaknesses:**

1: My main concern is that the analysis of the experiment is not inclusive; there is no clear comparison showing how LA improves reasoning other than vanilla CoT. Also missing ablation studies on individual components (e.g., impact of different logic rules, effectiveness of rule selection), and there is no detailed analysis of computational overhead or latency introduced by the framework and lack of error analysis showing standard failure modes

2: Insufficient details about prompt design choices and development process;  Limited explanation of how the framework ensures consistent rule application; Lack of engineering details about reproducing the results as well as the sensitivity to the prompts, which is  especially concerning given the contribution of the work is heavily engineered based

3: Novelty concern; heavily relying on GPT-4 for generation and evaluation creates potential circular reasoning which raises the questions on how can this be helpful in real world, especially given the improvement is not very significant.  Also, this seems to be limited to propositional logic without a clear path to handling more complex logic types, which raises questions about the generalizability of the method.

**Questions:**

1: How does the computational cost and efficiency scale?

2: What specific improvements in reasoning quality can be attributed to the framework versus simply better prompting? how do you justify such improvement quantitatively?

3: How were the prompting strategies developed, and what alternatives were considered?

4: How does the framework perform on more complex logical reasoning (Multi-gop, open ended,etc) tasks not covered in the current evaluation?

---

### Official Review · Reviewer_UNWT · 2024-10-29

**Soundness:** 1
**Presentation:** 1
**Contribution:** 1
**Rating:** 1
**Confidence:** 5

**Summary:**

The paper claims to propose a framework called Logic Agent which helps LLMs to choose from various propositional logic rules for reasoning. As the result,  LA  + LLM outperforms few shot chain-of-thought reasoning across various LLM models of different sizes.

**Strengths:**

Improving ability  for reasoning for LLMs is a very important field of research.

**Weaknesses:**

The paper promises a lot in introduction and in the introduction; however it unfortunately runs short in implementing them. Main weaknesses are as follows:

- It is not clear what is the "framework", it would really help to have an illustrative figure. Figure 1 and Figure 2 are more like a snapshot and the examples there are not really helpful (for instance Option A when we don't know other options, also many rules are based on absolute contrapositive, so  its not clear how they are useful.)

- Exposition lacks careful writing both on defining the formal reasoning tasks and the propositional logic component. (E.g., formal semantics and parsing side is purely verbal in terms of natural language.) Given all these paper is not technically substantiated.

- Important models such as Claude  and o1 is missing (although this is on the relatively minor side, since maybe the latter was not available by then).

- Besides, why the quantifiers if you are working with propositional logic. And if so, then it should be mentioned that these are QBF sentences. There are a few typos: liu et al.  and also the transitivity double arrow  technically does not  work.

-It is not clear whether the presented results are statistically valid.  For instance, how many shot  CoT has been done?

-Major related work  and comparison to other works as benchmarks are  missing e.g., works of Subbarao Kambhampati.

**Questions:**

Many of the rules are special case of contraposition (e.g., upper contrary law). How does including them additionally helps?

---

### Official Review · Reviewer_mQfq · 2024-10-31

**Soundness:** 4
**Presentation:** 2
**Contribution:** 2
**Rating:** 5
**Confidence:** 3

**Summary:**

The paper introduces a framework LA designed to enhance logical reasoning capabilities of LLMs by integrating propositional logic rules and deduction functions directly into the model's reasoning process. They use LLM as agents which performs the reasoning over the desired task. This work achieves this by converting natural language expression into the more concrete formal logic which helps in deciding chain of the reasoning to get the final answers. This work complement the chain of thought based model by introducing rule following and validate the logical inference.  Logic agent guide LLMs through predefined logical functions, which transform natural language inputs into logical forms that the model can interpret and apply with accuracy across various reasoning tasks. The work evaluates performance across different models, including GPT based and open-source LLMs (Mistral and LLAMA), showing that it significantly enhances accuracy in tasks that demand rigorous logical reasoning.

**Strengths:**

Verification of logical chain validity and use of symbolic reasoning model is compelling for the task requiring reasoning. Idea of making LLM as a agent for decision making makes sense as LLM are now widely used for tool selection and function calling. Presentation wise, I really liked the introduction part of the paper which makes compelling argument for the motivation of the work.
1. Presents a unique approach to improving logical reasoning in LLMs by transforming them into rule-guided agents.

2. Offers a promising alternative to existing Chain-of-Thought prompting techniques.

3. Comprehensive evaluation over multiple models and datasets of varying difficulty level.

4. The potential applications for Logic Agent in areas requiring logical rigor, such as legal reasoning and complex decision-making, make it a valuable development.

**Weaknesses:**

I really liked the introduction and related work section of the paper, which made very compelling premise for the logic agents. However section explaining the logic agents were not clear, apart from the Fig.2, section explaining them were not clear to me. Author(s) should put an effort so that readers who are not very deep and expert into this topic can benefit from this work. This requires through revision of the section 3 of the paper.  Some pointers, author say in 3.3 "we prompt LLM to discern and decide upon the most appropriate rule to apply...", but how this is done is missing from the papers, examples showing this will be helpful. I found paper to be bit rushed.
1. As this work is based on domain of prompting, it would be great to show how this look like, either in the paper or appendix. E.g., CoT paper does it quite nicely (for inspiration).
2. Appendix is cross referenced and it is written that code is available, but I found both to be missing.
3. How LLM are used as automated decision makers can be explained in details as this is a good innovation point but not explained properly.
4. Section 5.1 about parsers, how parsers prompting work is missing and makes hard to follow. It is written as a passing by argument but not explained. So nice opportunity is missed.

**Questions:**

1. Section 5.1 about parsers, details about default parsers is missing. Does it mean that LA agent is a default parsers?
2. In sec. 5.1 Author(s) wrote "To harness GPT-4’s parsing capabilities, we crafted specific prompts aimed at guiding the model to
translate natural language statements into propositional logic forms", how did you achieved that?
3. In results table 2, are there other SOTA model available, if yes why not compared against or can be shown in appendix?

---

### Official Review · Reviewer_PsbF · 2024-11-01

**Soundness:** 3
**Presentation:** 2
**Contribution:** 2
**Rating:** 5
**Confidence:** 2

**Summary:**

The authors present a Logic Agent (LA) framework for providing LLMs with a logical reasoning toolbox that can be used to assist in responding to queries requiring complex reasoning. They present a number of logical reasoning problems encountered by LLMs, and focus on contrapositive reasoning as an exemplar, something that LLMs have been shown to perform poorly on. This method can be applied generally to a wide variety of contexts and the authors present results on 7 different datasets from 3 task categories: Multiple choice reading comprehension, natural language inference and true of false questions, and on 5 different LLM backbones. The general approach requires the encoding of the context and query into propositional logic (with the addition of existential and universal qualifiers,) and the application of reasoning on these encodings using inference rules. Results show that their approach outperforms base LLMs (without LA) and chain of thought prompting (CoT) on the majority of tests with some cases being substantial improvement.

**Strengths:**

The overarching idea is a good one, to find a general way to encode context and query as propositions and to use common rules to help reason about these and avoid the LLMs making logic errors in their responses. The realisation of this into a workable scheme is pretty good too. This is evidenced by the general applicability of the approach and the fact that there are consistent improvements over other methods. There is a broad set of datasets and models being tested with a meaningful baseline and an ablation study.

**Weaknesses:**

The paper could be a little clearer in some parts. There is a little vagueness about how the method is applied. There are also some details that could be explained in a bit more detail. For instance, the authors state that the context is turned into propositional logic but that the forall and exists quantifiers are used (which suggests a first order-like encoding). It also wasn't clear how the logic generating step was developed to encode the context and what choices had been explored in achieving this. Finally, the writeup of the ablation study is very unclear and I am not sure what has been done here.

On another note, the analysis of the results felt somewhat oversold. There was a "significant leap" which was "particularly evident in datasets requiring complex reasoning" but this didn't really reflect the results. Some of the results were better by a margin for the LA over the other methods but this seemed less systematic than the authors were claiming. It would also help to identify where the method wasn't working so well and perhaps hypothesize about why that it. In short, it would benefit from a more objective analysis. On a related note, I also felt a bit like things were being oversold, particularly in the second half of the paper, with phrasing like: "capability to significantly refine", "firmly establishes LA as a transformative approach", "elucidates its broad applicability and effectiveness", "enhance the versatility" and so on. This is somewhat reminiscent of auto-generated text.

There  are some little niggling presentation errors that could do with clearing up too. For instance \citet seems to have been used throughout rather than \cite. And on page 8 there is a hanging reference ??

**Questions:**

How are `Variable` and the `forall` and `exists` quantifiers used within an otherwise propositional framework?

How are the LLMs guided to produce the guided generation? Are they fed the rules as text and asked to apply them? Or do you generate them functionally and feed to the LLM as additional context?

What happens if there is no clear encoding of the context into any usable atoms or implication rules? Is the LLM explicitly instructed in how to deal with that?

Section 5.1 seems to suggest that there was some default parser used as logic parser throughout. Is that right? Or was the corresponding LLM model used as a logic parser for its own pipeline? If the latter, then why don't we see the logic parsing results for the other models shown, just GPT-4?

How the parsing results in table 3 calculated? Do you have a parsing ground-truth for the data? If so, how was that established and how large is it?

In the ablation study how is the Parsed Logic method applied? Why have you chosen GPT-3.5 in this case?

---

### Note · Authors · 2024-11-23

I have read and agree with the venue's withdrawal policy on behalf of myself and my co-authors.